# Association between Perfluoroalkyl and Polyfluoroalkyl Substances and Women’s Infertility, NHANES 2013–2016

**DOI:** 10.3390/ijerph192215348

**Published:** 2022-11-20

**Authors:** Yuxuan Tan, Zurui Zeng, Huanzhu Liang, Xueqiong Weng, Huojie Yao, Yingyin Fu, Yexin Li, Jingmin Chen, Xiangcai Wei, Chunxia Jing

**Affiliations:** 1Department of Preventive Medicine and Public Health, School of Medicine, Jinan University, No. 601 Huangpu Ave West, Guangzhou 510632, China; 2Guangdong Women and Children Hospital, Guangzhou Medical University, Guangzhou 510632, China; 3Guangzhou Center for Disease Control and Prevention, Guangzhou 510440, China; 4Guangdong Key Laboratory of Environmental Exposure and Health, Jinan University, Guangzhou 510632, China

**Keywords:** PFAS, infertility, mixed effect, generalized linear model (GLM), generalized additive models (GAM), Bayesian kernel machine regression (BKMR)

## Abstract

Perfluoroalkyl and polyfluoroalkyl substances (PFASs) are widely used in consumer products. However, the role of PFAS in infertility is still poorly understood. A total of 788 women from the 2013–2016 nationally representative NHANES were included to explore the association between PFAS exposure and self-reported infertility. Six PFAS, including PFDE, PFNA, PFHxS, n-PFOA, n-PFOS, and Sm-PFOS, were detected by online SPE-HPLC-TIS-MS/MS. We used the generalized linear regression model (GLM), generalized additive models (GAM), and Bayesian kernel machine regression (BKMR) to assess the single effects, non-linear relationships, and mixed effects on women’s infertility, respectively. The prevalence of self-reported infertility was 15.54% in this study. In GLM, n-PFOA showed a negative association with self-reported infertility in women for the Q3 (OR: 0.396, 95% CI: 0.119, 0.788) and Q4 (OR: 0.380, 95% CI: 0.172–0.842) compared with Q1 (*p* for trend = 0.013). A negative trend was also observed in n-PFOS and ∑PFOS (*p* for trend < 0.05). In GAM, a non-linear relationship was revealed in Sm-PFOS, which exhibits a U-shaped relationship. The BKMR model indicated that there might be a joint effect between PFAS and women’s infertility, to which PFNA contributed the highest effect (PIP = 0.435). Moreover, age stratification analysis showed a different dose–response curve in under and above 35 years old. Women under the age of 35 have a more noticeable U-shaped relationship with infertility. Therefore, the relatively low level of mixed PFAS exposure was negatively associated with self-reported infertility in women in general, and the impact of PFAS on infertility may vary among women of different age groups. Further studies are needed to determine the etiological relationship.

## 1. Introduction

Infertility is a common reproductive disease, with a prevalence of 9% to 18% in the world’s general population, which involves about 15% of couples of childbearing age [1,2]. Women are more likely to suffer from fertility problems than men [3], and 1.5 million women in the United States had infertility from 2006 to 2010 [4]. These women may have harmful effects due to their infertility, such as societal repercussions, personal suffering, mood disorders [5,6,7], and sexual dysfunction [8,9].

Poly- and perfluoroalkyl substances (PFASs) belong to a family of highly fluorinated aliphatic compounds. Due to their hydrophobic and oleophobic properties, they are widely used in consumer products such as disposable food packaging, cookware, outdoor gear, furniture, and carpets [10]. PFAS was detectable in the blood of virtually all Americans (98%) according to a report by Centers for Disease Control and Prevention (CDC) [11]. Exposure to a high level of PFAS was associated with several reproductive health issues in women, including menarche delaying, menstrual cycle disorders, early menopause, premature ovarian failure, and dysregulation of circulating steroid homeostasis [12,13,14,15]. Experimental studies have shown that PFAS (2.0 to 17.5 ng/g feed in mice, 0.1 to 0.5 μM in zebra fish) has estrogenic properties in vitro and can adversely affect the reproductive system of experimental animals by disrupting the function of nuclear hormone receptors, interfering with steroid production, and changing the expression of endocrine-related genes [16,17,18]. Animal experiments also revealed that PFAS could cause reproductive damage in mice, pigs, cattle, and other mammals [19,20,21,22,23]. However, the general population’s exposure to environmental PFAS is usually lower than in animal experiments. The long-chain, legacy PFAS such as perfluorooctane sulfonic acid (PFOS) and perfluorooctanoic acid (PFOA) have long half-lives and may persist in the human body [24]. The arithmetical mean in individual apparent half-lives was estimated to be 5.0 years for PFOA and 6.5 years for PFOS [25]. It is more meaningful to conduct research on PFAS with low exposure levels in the general population. Meanwhile, the association of mixed PFAS exposure with other health impacts, such as cognitive function and persistent infections, has been demonstrated in previous studies [26,27], and the potential interaction between PFAS is also of great interest. However, the association of mixed PFAS exposure with women’s infertility has not been explored. Therefore, we aimed to investigate the relationship between PFAS and women’s infertility using a large and representative sample from the National Health and Nutrition Examination Survey (NHANES) 2013–2016 data. We also use three models, including generalized linear regression model (GLM), generalized additive models (GAM), and Bayesian kernel machine regression (BKMR) to explore the effects of single PFAS, non-linear relationships, and mixed PFAS exposure on women’s infertility.

## 2. Method

### 2.1. Study Design and Population

We selected the study population from the 2 cycles of NHANES (the year 2013–2014, 2015–2016), a cross-sectional, multistage probability sample representative of adults’ and children’s health and nutritional status in the United States [28]. The survey contains separated projects of interviews, physical examinations, and laboratory tests designed with stratified samples. The National Center for Health Statistics (NCHS) Research Ethics Review Board approved all study protocols, and all participants provided written informed consent.

We screened a total of 20,135 participants in the study. We excluded men (*n* = 9887), women without reproductive health-related data (*n* = 6525), and two-thirds of the participants were not sampled due to the serum PFAS concentration being measured in a one-third subsample of persons 12 years and over (*n* = 2567). We excluded 368 participants based on age (under 20) and pregnancy status (being pregnant). Finally, a total of 788 females were included in this study. Figure 1 shows the data integration process.

### 2.2. PFAS Measurement

In each survey cycle, a randomly selected one-third of participants over 12 years of age measured PFAS levels in serum. The measurement method of serum PFAS was described in previous studies [29]. According to the NHANES standard, we used the limit of detection (LOD) divided by the square root of two to replace the values below the LOD. Six PFAS with a detection rate over 65% in the 2013–2016 cycle were analyzed, including perfluorohexane sulfonic acid (PFHxS), pefluorodecanoic acid (PFDE), perfluorononanoic acid (PFNA), n-perfluorooctanoic acid (n-PFOA), n-perfluorooctane sulfonic acid (n-PFOS), and Sm- perfluorooctane sulfonic acid (Sm-PFOS). We also summed the Sm-PFOS and n-PFOS as the ∑PFOS according to the previous studies to assess the exposure of total PFOS [27,30].

### 2.3. Infertility Data

Self-reported infertility data were from NHANES reproductive health questionnaire (RHQ), and survey data were collected at all study sites by well-trained personnel following standardized procedures [31]. Briefly, the participants were asked two infertility-related questions. Firstly, they were asked: “Tried for a year to become pregnant?” and, “Have you ever been to a doctor or other medical provider because you have/she has been unable to become pregnant?” Those who answered these questions were enrolled in the study. One of these two questions answered “Yes” defined “ever infertile”, and both answered “No” described “fertility”. No response was considered missing [32].

### 2.4. Covariates

We identified sociodemographic, lifestyle, and survey-specific factors as covariates that could potentially bias the associations of PFAS exposures with infertility. In our analyses, age [33,34], race/ethnicity [35], body mass index (BMI) [36,37], family poverty income ratio (PIR), education level, physical activity, smoking status, alcohol drinking, and marital status were potential confounders based on the previous studies [32]. We also considered reproductive factors, including the age of menarche and reproductive history (any prior pregnancy), to control for bias in PFAS measurements due to pregnancy [38]. All these variables were extracted from NHANES questionnaires and laboratory measurements.

### 2.5. Statistical Analysis

Descriptive statistical analyses were applied to evaluate the demographic characteristics and self-reported infertility. Continuous variables were presented as medians with interquartile ranges (IQRs), and categorical variables were displayed as numbers (%). The baseline of characteristics of the infertility status was compared using the Mann–Whitney U test for continuous variables and the Wilcoxon rank-sum test for categorical variables. We simultaneously calculated and compared the serum PFAS concentrations in the baseline characteristics.

In generalized conditions, humans are exposed to several PFAS contemporaneously. Due to the skewed distribution of serum PFAS in the general population, we performed a log2 transformation for all PFAS. We applied three different methods to determine the impact of single, non-linear, and mixed PFAS exposure.

#### 2.5.1. Statistical Method 1: Generalized Linear Regression Model (GLM)

We performed the statistical analysis using four-year subsample B weights and strata variables for studies as required by the CDC analytical guidelines [39]. Multiple linear regression was used to evaluate the relationships between serum PFAS and self-reported infertility individually. Serum PFAS exposure levels were divided into 4 quantiles in GLM modeling, as most recent studies have reported [26,32,40]. In Model 1, we did not adjust any covariate. Model 2 included age, BMI, race, education, PIR, physical activity, smoking status, serum creatinine, alcohol drinking, stroke, marital status, age of menarche, and reproductive history.

#### 2.5.2. Statistical Method 2: Generalized Additive Model (GAM)

Considering the GLM method might not be adequately fitted in potential non-linearity relationship, we applied a generalized additive model (GAM) to reveal whether there was a nonlinear relationship between serum PFAS exposure and self-reported infertility. GAM is an extension of the GLM, which allows the evaluation of the non-linear relationship between the outcome and the predictors. It provides insight into the relationship between response variables and explanatory variables [41]. The estimated degree of freedom (EDF) was used to represent the complexity of the smooth. When the EDF is greater than 1, it is considered that there is a nonlinear relationship, with higher EDFs describing wigglier curves. We used ANOVA to test whether the smoothing term is statistically significant. In the GAM model, and all covariates were adjusted to control the basis.

#### 2.5.3. Statistical Method 3: Bayesian Kernel Machine Regression (BKMR)

To examine associations between serum PFAS and self-reported infertility, we performed Bayesian kernel machine regression (BKMR) to investigate the single and mixed exposure. BKMR estimates the model via Bayesian inference to account for uncertainty due to evaluating a high-dimensional set of directions and multiple-testing penalty [42]. Briefly, BMKR models the non-linear function using a Gaussian process model with a radial basis function (RBF) kernel, and measures each PFAS individual contribution by locating a spike-and-slab before the pollutant components [43]; 50,000 iterations were conducted for the BKMR models with all covariates adjusted. All six PFAS were included using the variable selection option to assess the individual posterior inclusion probability (PIPs). A cumulative effect was calculated by fitting the predictors at 25th and 50th percentiles, respectively, to reveal the different reference point selecting scenarios. As noted by Bobb et al., we also conducted a series of sensitivity analyses, including changing the prior distribution and adjusting the smoothness of the kernel to assess the robustness of our BKMR model. The information about BKMR and sensitivity analyses were described in Appendix A.

### 2.6. Stratified Analyses

Numerous studies have shown that age has a significant effect on women’s fertility [33,44,45]. We performed a stratified analysis to investigate the association between women’s infertility and serum PFAS in different age groups. We separated the different ages into two subgroups by 35 years of age (median age of present study) and performed both GLM and GAM methods for the two groups separately to investigate the impact of serum PFAS on women’s infertility in different age groups.

GAM and BKMR do not currently support adjustments for clustered sampling schemes, so NHANES weights and strata variables were not included in these models. All analyses used the Stata software (Version 17, Stata Corp, College Station, TX, USA) and R packages (R Development Core Team, https://cran.r-project.org/ (accessed on 27 April 2022)). Statistical significance was set at *p* < 0.05.

## 3. Results

### 3.1. Population Characteristics

Appendix A presents the participants’ general characteristics (*n* = 788). The mean age was 35.48 years, and nearly 70% of participants were overweight or obese (BMI > 24.9). Women with infertility were older, more educated, and with a higher proportion married than in the control group. There were no significant differences in serum cotinine, drinking status, BMI, family PIR, physical activity, age of menarche, and ever pregnant.

### 3.2. Distribution and Correlation of Serum PFAS

Table 1 summarizes the distributions of the PFAS and the percent that were higher than the LOD. 6 PFAS or their congeners are above LOD among the 65% of participants. The concentration of Sm-PFOS is lower in the infertility group (*p* < 0.05). There was no statistical difference in other serum PFAS concentrations between the two groups (*p* > 0.05).

The Spearman correlations among the six log-transformed PFAS are shown in Figure 2. PFDE was strongly correlated with PFNA (r = 0.72, *p* < 0.01) and moderately correlated with PFHxS (r = 0.28, *p* < 0.05). PFHxS was also correlated with PFNA (r = 0.48, *p* < 0.01).

### 3.3. Using GLM to Evaluate Single PFAS Exposure

In the full adjusted GLM model, n-PFOA was negatively associated with women’s infertility in the Q3 [OR (95% CI): 0.396 (0.199, 0.788)] and Q4 [OR (95% CI): 0.380 (0.172, 0.842)] compared with Q1 (Table 2). A negative association was also found in PFNA for the Q3 [OR (95% CI): 0.430 (0.214, 0.860)], while *p* for trend showed no significance (*p*-*t* = 0.098). n-PFOS and ∑PFOS also showed a negative trend with women’s infertility (both *p*-*t* = 0.032).

### 3.4. The Association between Serum PFAS and Self-Reported Infertility by the GAM

Figure 3 showed the trend of each PFAS exposures. In the GAM analysis, a linear negative association was found in n-PFOA and self-reported infertility (EDF = 1, *p* < 0.01 **), while Sm-PFOS showed a “U-shaped” relationship (EDF = 2.975, *p* < 0.05 *). There is a potential non-linear relationship between other PFAS and women’s infertility (EDF > 1, *p* > 0.05). In total PFAS, ∑PFOS also showed a potential “U-shape” with women’s infertility (EDF = 3.673, *p* > 0.05) (Appendix A). Appendix A shows the details of GAM modeling results.

### 3.5. The Association between Serum PFAS and Self-Reported Infertility by the BKMR Model

BKMR revealed a linear association between individual PFAS and infertility similar to the results of GAM (Appendix A). Posterior inclusion probabilities (PIP) in BKMR were shown in Appendix A, in which PFNA (PIP = 0.435) played the most essential role in overall effects. The PFAS mixtures showed a negative association with women’s infertility in the BKMR model (Figure 4A,B). A negative trend of self-reported infertility risk and the combined PFAS exposure was evident when co-exposure exceeded the 25th percentile (Figure 4A). There was no evidence for interactions between PFAS (Appendix A).

In BKMR sensitivity analysis, although the overall estimate value of three models and original model were quite different, the trend of the overall effect was robust (Appendix A).

### 3.6. Stratified Analyses

Interestingly, the trend between PFAS mixed exposure and women’s infertility differed between the under 35-year-old and over 35-year-old groups. At age under 35, a “J-shaped” or “U-shaped” association was observed in PFDE, PFNA, n-PFOS, and Sm-PFOS and women’s infertility. There were only negative trends observed over 35 (Figure 5). Appendix A show the baseline of two age-stratified groups. The multivariate linear results stratified by age showed that for women younger than 35 years, there is no an association between n-PFOA and self-reported infertility (*p* > 0.05), whereas n-PFOA for the Q4 [OR (95% CI): 0.33 (0.12, 0.92)] were significantly associated with infertility in women older than 35 years (Appendix A).

## 4. Discussion

This is the first study to explore the relationship between the mixed PFAS exposure and women’s infertility in the representative general U.S. population. We found a non-linear relationship between the prevalence of self-reported infertility among women and serum concentrations of PFAS (Figure 3), suggesting that the effect of PFAS on fertility might depend on exposure levels and/or different subtypes.

PFAS are common endocrine disrupting chemicals (EDCs) detected in 99–100% of pregnant women [46]. Non-monotonic dose responses (NMDR) in EDCs were widely observed, and a curve slope changes direction within the range of tested doses [47,48]. As typical EDCs, PFAS has been reported the effect of NMDR. Mancini et al. revealed an inverse U-shaped association between PFOA dietary exposure and the risk of developing type 2 diabetes [49]. A Swedish cohort study also reported a NMDR relationship between PFOS and overweight/obesity in children [50]. A U-shaped association between PFOA and cognitive function in older adults was identified [51]. In addition, PFAS followed a prevalent inverted U-shaped distribution across patients in declining stages of glomerular function [52].

Although it is challenging to present plausible explanations with limited evidence, several potential mechanisms were proposed to explain the NMDR effects of PFAS, including estrogen-like effects, low-dose stimulation effects, and cytotoxicity. PFAS may promote modifications of endogenous hormone regulation in humans and in wildlife [19,53,54,55]. PFAS showed weak estrogenic effects in animal experiments, which manifested in increased estrogen and progesterone concentrations or mimicked the effect of endogenous estrogen [56,57,58]. PFAS can modulate the endocrine system by up- or downregulation of the expression of proteins responsible for cholesterol transport and ovarian steroidogenesis [53,59,60]. PFOA-treated ovary-intact mice had significantly increased serum progesterone (P) levels [56]. Cytological findings suggest that PFOS inhibits the conversion of P to testosterone by inhibiting CYP17 [61]. PFOA, PFNA, PFDA, and PFOS are all efficiently combined with estrogen receptors alpha (*ERα*) in different species [57]. Meanwhile, PFOS induced E2 production and reduced testosterone (T) production in a concentration-dependent manner in the H295R cells [61]. Previous studies have confirmed that estradiol/progesterone and its substitution could improve pregnancy rates in the luteal phase [62,63,64]. Hence, we speculate that exposure to PFAS in a specific range of concentrations might benefit fertility.

Our result presented a negative trend of exposure to low-dose PFAS on women’s infertility, which has also been observed for a wide variety of EDCs (e.g., bisphenol A (BPA), phthalates) [47,65,66,67]. The median and IQR of ΣPFOA and ΣPFOS in our research (NHANES 2014–2016) were 1.27 (0.77 to 22.59) ng/mL and 3.30 (1.20 to 5.30) ng/mL, respectively, which were lower than those studies that reported positive associations (Appendix A) [13,68,69,70]. Additionally, studies that reported lower concentrations than our study did not find associations between PFAS and infertility [69]. Low levels of EDCs exposure may cause a hormesis effect [71,72], which describes a biphasic dose response to an environmental agent with a low-dose stimulation showing beneficial effects and a high-dose stimulation leading inhibitory or toxic effects [73,74]. A low concentration (33 ng/L) of ethinylestradiol (EE2) could induce hormesis (immune enhancement), enable adaptation (restored reproduction), and even boost fish resistance to the bacterial challenges after abatement of EE2. As our previous study, low-dose PFOA and PFOS might present a hormesis-effect, which shows a positive trend on cognitive function [26]. Evidence on the hormesis effects of PFAS on reproductive function is currently lacking, and we encourage researchers to explore this in greater depth longitudinally.

Although it is well known that age plays a vital role in fertility [75,76], the differences between serum PFAS levels and infertility in the age-stratified analysis are of great interest, especially in PFDE, PFNA, n-PFOS, and Sm-PFOS (Figure 5). Women enter perimenopause between the ages of 35 and 50, hormone levels change significantly [77]. We consider that sex hormone levels in young women are relatively stable and even low-level PFAS exposure might cause more pronounced physiological changes. Moreover, previous studies have shown that serum PFAS concentration appears to be age-specific [55,78]. Women of younger age have lower concentrations of PFAS [55], and due to the short exposure period and vigorous metabolism, PFAS is relatively easier to exclude. As they age, PFAS accumulates in their bodies, resulting in relatively higher serum PFAS levels in older women [10]. Therefore, we could hypothesize that the accumulation of PFAS in perimenopausal women might occasionally result in a negative trend between PFAS and infertility, making the reproductive toxicity in same exposure levels of PFAS less sensitive in perimenopausal women than in non-perimenopausal women. However, our results still need to be interpreted with caution.

Previous studies showed inconsistent associations between PFAS exposures and women’s infertility and infertility-related diseases [68,70,79]. A case-control study (*n* = 240) in China showed that exposure to PFOA and PFOS increased the risk of premature ovarian insufficiency (POI) at the age of around 30 (Mean ± SD: 28.9 ± 5.6). A case-control study (*n* = 97) in Australia showed the links between PFAS exposures and increased risk of infertility factors like endometriosis and POS [79]. Women with higher PFOA (≥4.20 ng/mL) and PFNA (≥1.50 ng/mL) serum levels were less likely to become pregnant than those with lower levels in a prospective study [80]. PFOA (≥3.91 ng/mL) and PFOS (≥26.1 ng/mL) exposure may reduce fecundity in Danish women [81]. PFNA, PFOA, and PFOS were also associated with endometriosis in a study of U.S. women aged around 20 to 50 years from NHANES 2003–2006 [82]. However, In a survey from Zhejiang, China (*n* = 335), relatively lower levels of plasma PFAS (PFHpA, PFHxS, PFNA) were inversely associated with endometriosis-related infertility [70]. Our findings add the negative association of PFAS with women’s infertility to the current literature, and more academic exploration should be made to clarify further reasons.

Three statistical methods explored the relationship between PFAS exposure and women’s infertility in different dimensions, which is very important for interpreting the consistency of the results. The GLM method was generally used in traditional health impact and risk investigation. GLM revealed an inverse association between individual PFAS and women’s infertility, but it cannot identify the NMDR and the overall effect of mixed exposure. Therefore, this study employed a non-linear model, GAM, for further analyses. GAM is widely used in the non-linear exploration of exposure and health outcomes due to its flexible fitting. The non-linear relationship was detected in Sm-PFOS (EDF > 1, *p* < 0.05), and potential non-linear relationships were also revealed in PFDE, PFHxS, PFNA, n-PFOS. To assess the joint effect of PFAS exposure, the BKMR model can evaluate the relationships between mixed exposures and health outcomes, allowing for the nonlinear and non-additive exposure-response function. We found that PFNA have the highest PIP in mixed exposures. Sensitive analysis of BKMR indicated a negative trend of the overall effect, which proved the stability of negative associations. Three statistical methods examined the relationship between PFAS and women’s infertility from different dimensions, which validated the results’ stability and reduced the possibility of accidental errors. However, the effect of PFAS on women’s infertility cannot be fully explained, and more in vitro/in vivo experiments and population experiments are needed to confirm their relationship.

This study has some obvious advantages. First, this is the first study to determine the impact of PFAS mixture exposure on U.S. women’s general infertility, which may provide new perspectives on infertility. Secondly, we used three different statistical methods to assess the relationship between PFAS and infertility. Notably, we revealed a “U-shaped” dose–response relationship in this research, further supporting the hypothesis of a non-linear relationship between low-dose exposure to PFAS and infertility. However, we cannot conclude the causal relationship between PFAS exposure and women’s infertility due to the cross-sectional study design. One-time measurement of serum PFAS levels is not representative of the long-term exposure of this population, and self-reported infertility is also not representative of obstetric examination results. Even if we included reproductive history as a covariate, the impact of pregnancy on women’s serum PFAS concentrations could not be ruled out. Meanwhile, the number of participants who had reported infertility is relatively low. We cannot conduct an age-stratified analysis in more specific age groups. Reverse causality might not be avoided. More research is needed.

## 5. Conclusions

We highlighted a controversial result that negative associations with PFAS and women’s infertility varied according to types of PFAS and age. GAM revealed a prevalent non-linear association between PFAS and women’s infertility. Mixed PFAS exposure might influence infertility negatively as revealed in the BKMR model. Our study indicated that further profound studies are needed to address the impact of low-dose PFAS exposure on women’s infertility. Future longitudinal studies are required to confirm the exact relationship between PFAS and women’s infertility and the basic mechanisms.

## Figures and Tables

**Figure 1 ijerph-19-15348-f001:**
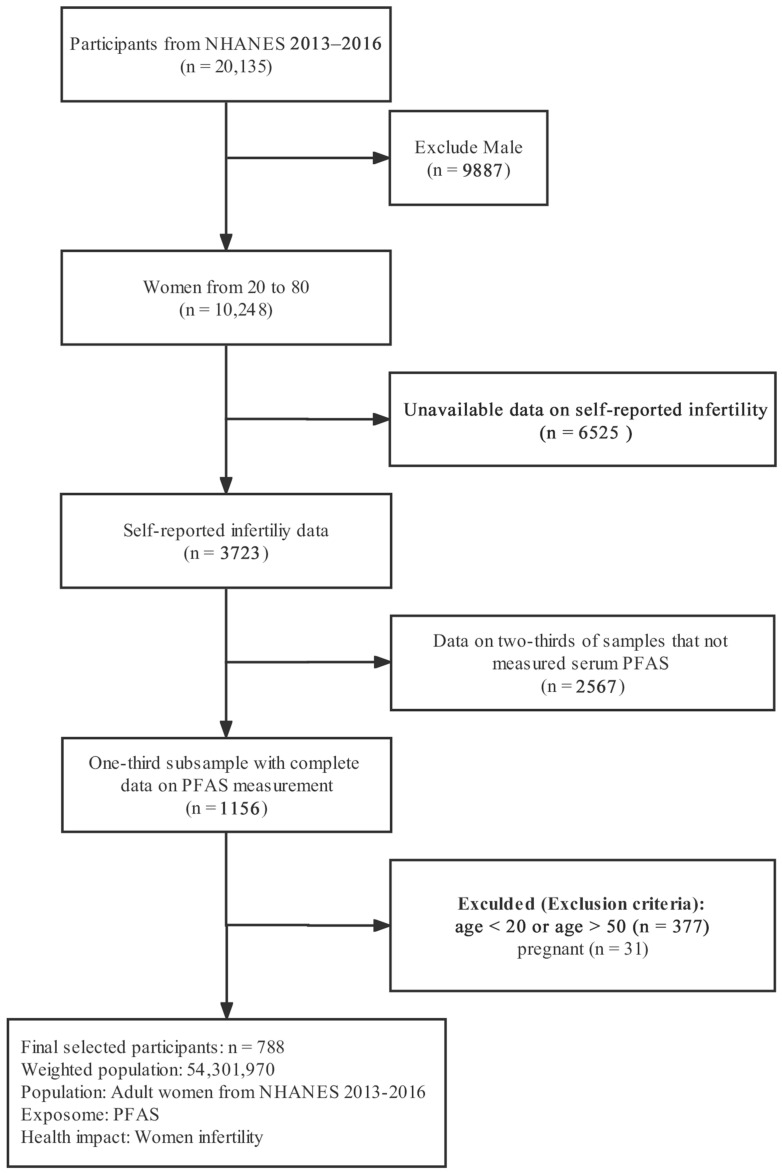
Flow chart of the selection of eligible participants, NHANES 2013–2016.

**Figure 2 ijerph-19-15348-f002:**
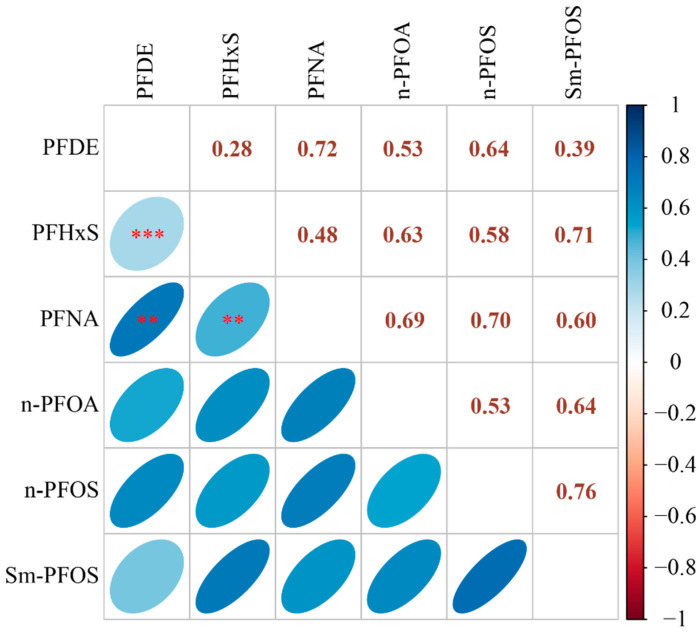
The correlation of the 6 PFAS. The blue color represents the positive correlation, and the red color represents the negative correlation. A darker color indicates a stronger correlation. Note: ** *p* < 0.01; *** *p* < 0.001.

**Figure 3 ijerph-19-15348-f003:**
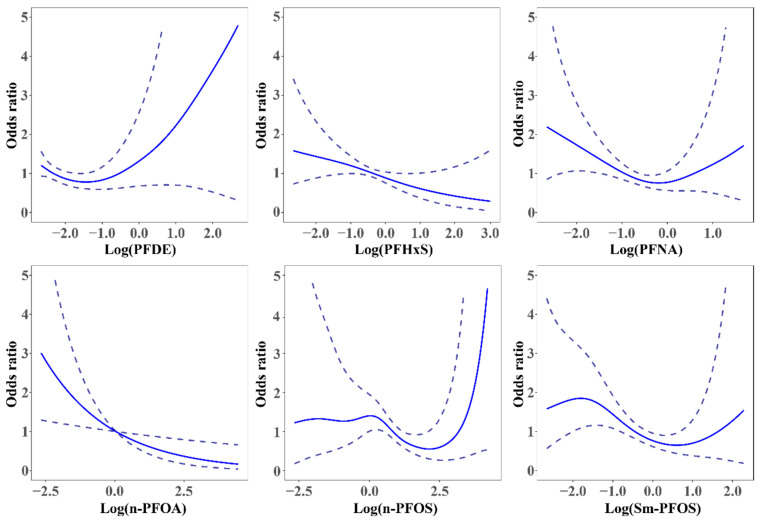
Effect of PFAS exposure and women’s infertility in the full adjusted multivariable GAM. The blue dotted line represented the 95% CI, and the blue solid line represented the estimate OR.

**Figure 4 ijerph-19-15348-f004:**
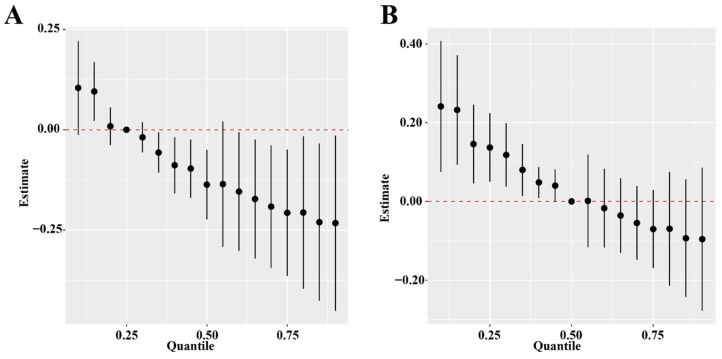
The overall effect of the point estimates and their 95% credible intervals (95% CrI) for the difference at various quantiles (ranging from 0.10 to 0.90). (**A**) Estimated value compared to fixing all PFAS concentrations at their 25th percentile and (**B**) estimated value compared to fixing all PFAS concentrations at their 50th percentiles.

**Figure 5 ijerph-19-15348-f005:**
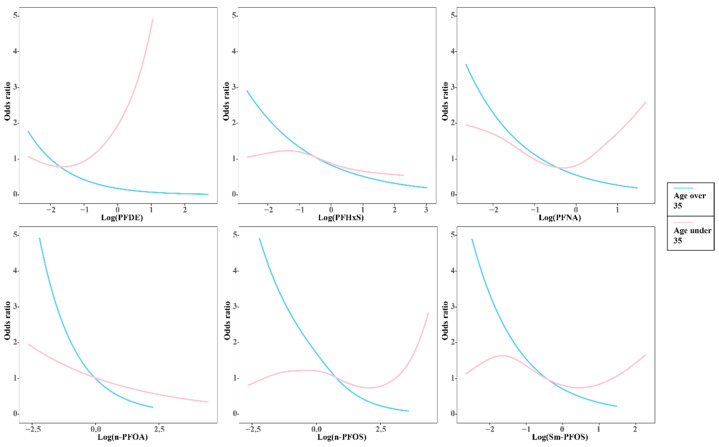
The subgroup analysis in full adjusted GAM. The trend of PFAS exposure and self-reported infertility was different between two groups.

**Table 1 ijerph-19-15348-t001:** Description of perfluoroalkyl levels among participants, NHANES 2013–2016 ^a^.

Exposure	LOD (ng/mL) ^b^	N (%) of Below LOD	Total	Infertility		*p*-Value
				No	Yes	
N			788	682	106	
Individual PFAS (ng/mL)						
PFDE, Median [IQR]	0.10	32.73%	0.10 [0.07, 0.20]	0.10 [0.07, 0.20]	0.10 [0.07, 0.20]	0.620
PFHxS, Median [IQR]	0.10	1.92%	0.60 [0.40, 1.02]	0.60 [0.40, 1.10]	0.60 [0.30, 0.80]	0.078
PFNA, Median [IQR]	0.10	1.72%	0.50 [0.30, 0.80]	0.50 [0.30, 0.80]	0.40 [0.30, 0.70]	0.184
n-PFOA, Median [IQR]	0.10	0.79%	1.10 [0.70, 1.60]	1.10 [0.70, 1.60]	0.90 [0.60, 1.50]	0.083
n-PFOS, Median [IQR]	0.10	0.72%	2.20 [1.30, 3.50]	2.20 [1.30, 3.58]	1.85 [1.30, 3.08]	0.066
Sm-PFOS, Median [IQR]	0.10	1.37%	0.70 [0.40, 1.10]	0.70 [0.40, 1.20]	0.55 [0.32, 1.10]	0.041 *
Total PFAS (ng/mL)						
∑PFOS, Median [IQR]	-	-	1.17 [0.77, 1.70]	1.17 [0.77, 1.70]	0.97 [0.67, 1.59]	0.081

^a^ Median (and interquartile range, IQR) are shown for the PFAS. ^b^ The limit of detection (LOD) was not available because total PFAS were calculated by its isomers. Note: * *p* < 0.05.

**Table 2 ijerph-19-15348-t002:** Association between PFAS exposure and women’s infertility using GLM.

PFAS	Quartile1	Model 1 OR (95% CI)	*p*-Value	Model 2 OR (95% CI)	*p*-Value
Individual PFAS					
PFDE	Quartile1	Ref.		Ref.	
	Quartile2	0.674 (0.297, 1.533)	0.335	0.738 (0.292, 1.862)	0.507
	Quartile3	0.582 (0.283, 1.198)	0.136	0.541 (0.232, 1.262)	0.149
	Quartile4	0.886 (0.463, 1.694)	0.705	0.776 (0.414, 1.453)	0.415
	*p*-*t*	0.429		0.236	
PFHxS	Quartile1	Ref.		Ref.	
	Quartile2	0.496 (0.198, 1.242)	0.129	0.442 (0.185, 1.054)	0.065
	Quartile3	1.151 (0.591, 2.241)	0.670	0.987 (0.481, 2.025)	0.97
	Quartile4	0.532 (0.253, 1.118)	0.093	0.532 (0.236, 1.199)	0.123
	*p*-*t*	0.295		0.337	
PFNA	Quartile1	Ref.		Ref.	
	Quartile2	0.682 (0.316, 1.474)	0.318	0.660 (0.297, 1.467)	0.297
	Quartile3	0.537 (0.252, 1.144)	0.104	0.430 (0.214, 0.860)	0.019 *
	Quartile4	0.650 (0.278, 1.520)	0.309	0.580 (0.252, 1.331)	0.190
	*p*-*t*	0.218		0.098	
n-PFOA	Quartile1	Ref.		Ref.	
	Quartile2	0.785 (0.471, 1.310)	0.342	0.664 (0.390, 1.131)	0.127
	Quartile3	0.509 (0.296, 0.877)	0.017 *	0.396 (0.199, 0.788)	0.010 *
	Quartile4	0.502 (0.240, 1.046)	0.065	0.380 (0.172, 0.842)	0.019 *
	*p*-*t*	0.035 *		0.013 *	
n-PFOS	Quartile1	Ref.		Ref.	
	Quartile2	1.440 (0.669, 3.102)	0.339	1.819 (0.930, 3.557)	0.079
	Quartile3	0.791 (0.367, 1.704)	0.537	0.773 (0.358, 1.670)	0.500
	Quartile4	0.660 (0.330 1.323)	0.232	0.589 (0.288, 1.204)	0.141
	*p*-*t*	0.111		0.032 *	
Sm-PFOS	Quartile1	Ref.		Ref.	
	Quartile2	0.816 (0.391, 1.699)	0.575	0.780 (0.342, 1.778)	0.543
	Quartile3	1.006 (0.502, 2.015)	0.986	0.861 (0.426, 1.739)	0.667
	Quartile4	0.648 (0.317, 1.325)	0.225	0.461 (0.200, 1.062)	0.068
	*p*-*t*	0.301		0.076	
Total PFAS					
∑PFOS	Quartile1	Ref.		Ref.	
	Quartile2	1.06 (0.59, 1.902)	0.841	1.303 (0.762, 2.229)	0.321
	Quartile3	0.563 (0.279, 1.138)	0.106	0.539 (0.261, 1.113)	0.092
	Quartile4	0.677 (0.348, 1.317)	0.240	0.557 (0.281, 1.104)	0.091
	*p*-*t*	0.127		0.032 *	

Note: * *p* < 0.05; OR, Odd ratio; CI, confidence interval; *p*-*t*, *p*-value for trend; Model 1: unadjusted model, Model 2: adjusted for all covariates.

## Data Availability

All data in the article can be downloaded for free in the NHANES database from https://www.cdc.gov/nchs/nhanes/ (accessed on 27 February 2022).

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
