# Peer review of "Association between Perfluoroalkyl and Polyfluoroalkyl Substances and Women’s Infertility, NHANES 2013–2016"

_ijerph, 2022, doi:10.3390/ijerph192215348_

Round 1

Reviewer 1 Report

This manuscript reports highly relevant research on potential health effects of PFAS. The scientific approach is sound but the presentation could be improved prior to publication.

1. The title should refer to "women's infertility" or "female infertility." The phrasing "women infertility" is awkward.

2. The sections describing the mathematical treatments and results are well written overall. The introduction contains several instances of awkward phrasing that should be revised. Some examples follow.

a.) Line 54 suggests that disposable food packaging is a property of PFAS rather than linking packaging and other examples to consumer products.

b.) Line 59: Replace "estrogen" with "estrogenic."

c.) Line 66: The full names for PFOS and PFOA are incorrect. The anion forms of the chemicals are given rather than the neutral molecule forms.

d.) Many of the references cited are outdated. As PFAS health effects is an active area of research, more references from within the last 5 years would be appropriate.

e.) Lines 58-59: Sex steroid hormones circulation is a normal biological function. Changes to circulation levels may be associated with PFAS levels, but circulation itself is not. The sentence should be revised for clarity.

f.) Lines 50-51: The sentence suggests that malignant tumors are caused by infertility. The authors should re-evaluate the phrasing of the sentence so that it is clear that malignant tumors may instead cause infertility.

3.  All abbreviations should be explained at first use in the text, including CDC, NCHS, GAM, GLM, and BKMR.

4. Unless the authors were involved with the analytical chemistry data acquisition (p. 4, lines 98-101), the sentence starting with "The Online solid phase extraction..." should be deleted. The sentence on line 101 is a sufficient description. As written, it appears that the authors performed the analytical chemistry.

5. Figure 2: The meaning of the red asterisks should be explained in the caption for the figure.

6.  Table 1: The data re nicely summarized in the text. Table 1 should be moved to a supplemental data section.

7. Table 2. What are the units for the limit of detection?

Author Response

Response to Reviewer 1 Comments

General comment: This manuscript reports highly relevant research on potential health effects of PFAS. The scientific approach is sound but th e presentation could be improved prior to publication.

Response: Thank you for the nice comments and critical suggestions that led to possible improvements in the current version of our manuscript. We have read these comments attentively and responses point-by-point below.

Point 1: The title should refer to "women's infertility" or "female infertility." The phrasing "women infertility" is awkward.

Response 1: Thank you for your comments. We have revised our title to “Association between perfluoroalkyl and polyfluoroalkyl sub-stances and women’s infertility, NHANES 2013-2016” and double-checked the whole manuscript to avoid these inappropriate expressions.

Point 2: The sections describing the mathematical treatments and results are well written overall. The introduction contains several instances of awkward phrasing that should be revised. Some examples follow a.) Line 54 suggests that disposable food packaging is a property of PFAS rather than linking packaging and other examples to consumer products. b.) Line 59: Replace "estrogen" with "estrogenic." c.) Line 66: The full names for PFOS and PFOA are incorrect. The anion forms of the chemicals are given rather than the neutral molecule forms. d.) Many of the references cited are outdated. As PFAS health effects is an active area of research, more references from within the last 5 years would be appropriate. e.) Lines 58-59: Sex steroid hormones circulation is a normal biological function. Changes to circulation levels may be associated with PFAS levels, but circulation itself is not. The sentence should be revised for clarity. f.) Lines 50-51: The sentence suggests that malignant tumors are caused by infertility. The authors should re-evaluate the phrasing of the sentence so that it is clear that malignant tumors may instead cause infertility.

Response 2: We deeply appreciate the reviewer for pointing out the inappropriate expressions in our manuscript. All the awkward phrases are fixed, and replied to point-by-point as below:

a). We have rewritten this sentence as follows: Due to their hydrophobic and oleophobic properties, they are widely used in consumer products such as disposable food packaging, cookware, outdoor gear, furniture, and carpets (page 2, lines 53 - 55).

b). Thank you. We have revised it (page 2, line 61).

c). The full names of PFOA (perfluorooctanoic acid) and PFOS (perfluorooctane sulfonic acid) in the manuscript were corrected according to PubChem (a database of chemical modules, https://pubchem.ncbi.nlm.nih.gov/) (page 2, line 67).

d). We have updated the references on the effects of PFAS on female reproductive health (page 2, lines 59). Some animal experiments and mechanism exploration references were also updated (page 2, lines 63).

e). The sentence has been rewritten into “Exposure to a high level of PFAS was associated with several reproductive health issues in women, including menarche delaying, menstrual cycle disorders, early menopause, premature ovarian failure, and dysregulation of circulating steroid homeostasis” (page 2, lines 57 - 59).

f). The association between malignancy and female infertility remains rare. To avoid ambiguity for our readers, we have removed the description that infertility may lead to malignancy (page 2, line 51).

Point 3: All abbreviations should be explained at first use in the text, including CDC, NCHS, GAM, GLM, and BKMR.

Response 3: Thank you for your comments. We have double-checked the manuscript and explained all abbreviations at the first use (page 2, line 56; lines 79 - 79; line 87).

Point 4: Unless the authors were involved with the analytical chemistry data acquisition (p. 4, lines 98-101), the sentence starting with "The Online solid phase extraction..." should be deleted. The sentence on line 101 is a sufficient description. As written, it appears that the authors performed the analytical chemistry.

Response 4: Thank you for your constructive comments. We have removed the improper expression of the analytical chemistry data source (page 3, lines 101 - 102).

Point 5: Figure 2: The meaning of the red asterisks should be explained in the caption for the figure.

Response 5: Thank you for your comments. The blue color represents the positive correlation, and the red color represents the negative correlation. A darker color indicates a stronger correlation. ** P < 0.01; ***P < 0.001. We have added these descriptions in the caption of Figure 2 (page 6, lines 211 - 213).

Point 6: Table 1: The data are nicely summarized in the text. Table 1 should be moved to a supplemental data section.

Response 6: Thank you for your suggestions. We have moved Table 1 to the supplementary files (supplementary files Table S1).

Point 7: Table 2. What are the units for the limit of detection?

Response 7: Thank you for your comments. The units for the LOD are “ng/ml”, and we have added this in Table 1.

Reviewer 2 Report

the manuscript is well presented and understandable. However, a few grammatical mistakes were detected and require amendment. Please refer to the attached commented manuscript for further information. 

Author Response

Response to Reviewer 2 Comments

Comments and Suggestions for Authors: The manuscript is well presented and understandable. However, a few grammatical mistakes were detected and require amendment. Please refer to the attached commented manuscript for further information.

Response: Thank you for making these constructive comments to improve the presentation and grammar of the paper. We have meticulously double-checked our manuscript and have necessary revised our manuscript according to your notes. All changes have been marked in red in the revised version of the manuscript.

Reviewer 3 Report

Word file attached

Author Response

Response to Reviewer 3 Comments

General comment: The study titled “Association between perfluoroalkyl and polyfluoroalkyl substances and women infertility, NHANES 2013-2016” by Tan et al. describes the intersection of PFAS environmental exposure and women’s reproductive health. It is an important topic on infertility that women are facing all around the world. The study is interesting, and the different models help to explore linear, non-linear, and mixed relationships. However, I am not confident how representative the samples are for the US population since it is a single point measurement of PFAS (mentioned by authors in the limitation section). Also, it is not very clear how the goal of this study is different from previous studies. In the discussion section, the authors do mention other studies with more than one PFAS measured and how the negative association is similar to other studies.

Response: Thank you for your critical and constructive comments which definitely contribute to improving our manuscript. We have studied the comments carefully and have revised the manuscript according to them. Accordingly, we have responded, point by point, to the comments as listed below.

In this study, we used single-measurement exposure data to represent the participants' PFAS exposure levels, which is a limitation. PFAS has a long half-life in humans[1]. Since PFAS exposure is routine in the population, a single measurement of serum PFAS concentrations can approximate recent exposures. This study focused on female self-reported infertility during the cross-sectional period, so a single measurement could represent general exposure.

In addition, we used the NHANES complex survey data for revealing the relationship between women’s infertility and PFAS exposure. NHANES is designed to be representative of the civilian, non-institutionalized resident population of the United States.  Even though only 788 participants’ PFAS exposure levels were measured in this study, we can still generalize the results of the weighted analysis to the entire U.S. population by adjusting the weights. The details were showed in the NHANES official website (https://wwwn.cdc.gov/nchs/nhanes/tutorials/module2.aspx). Moreover, all weighted analyses were conducted based on the NHANES analysis guide (https://wwwn.cdc.gov/nchs/nhanes/tutorials/default.aspx).

Although several population-base studies are exploring the relationship between PFAS exposure and women’s infertility[2-4], the conclusions are inconsistent. A study from China revealed the plasma concentration of perfluorododecanoic acid (PFDoA) was associated with a significantly increased risk of PCOS-related infertility[2]. Preconception exposure to 6:2 diPAP and PFHpA in women may potentially impair couple fecundity[4]. The study by Wang et al. suggests that most PFAS were not associated with early pregnancy loss[5]. Moreover, studies about the relationship between PFAS and female reproductive health are still relatively scarce globally, and the association between mixed PFAS exposure and women’s reproductive health in the U.S. population is rarely explored.

Therefore, it is necessary to explore these relationships in different populations and methods. It is the first study that used a nationally representative population sample from the U.S. to reveal the effects of single and mixed PFAS exposure on women’s infertility, which is worthy of being highlighted.

Specific comment 1: Introduction section has many good references to the topic. However, they are written in broad terms. For example, adding PFAS concentrations for References 21-23 and half-lives values for Reference 29 will make it more informative.

Response 1: Thank you for your comments. We have revised the introduction section of our manuscript to make it more detailed (page 2, lines 68 - 70; page 2, lines 60 - 61).

Specific comment 2: What are the other diseases that References 30 and 31 refer to?

Response 2: Weng et al. revealed a negative association between PFAS exposure and cognitive function in a population aged over 60 (reference 30). Bulka et al. suggested that PFAS might systematically increase susceptibility to persistent infections in the general population (reference 31). We have added these details in the revised version of our manuscript (page 2, line 72).

Specific comment 3: Figure 1 has incorrect wording in box 4 – Is it available or unavailable data that led to the n=6525 exclusion?

Response 3: Thank you for your comments. The participants without available fertility data were excluded from our study (n = 6525). We have revised figure 1 in our manuscript (page 3, line 97).

Specific comment 4: For section 2.3, is there a reference for your assumptions regarding “Yes” and “No” answers?

Response 4: Thank you for your comments. We have cited a related reference for definitions of the health outcome (page 4, page 119).

Specific comment 5: Table 2. Footnote “a” says mean and std dev are shown for PFAS. However, all the individual PFAS are tabulated with Median and IQR. Not clear what the footnote is referring to.

Response 5: Thank you for your comments. We are so sorry for the unclear description in the footnote. We have revised this in the manuscript (page 6, lines 203).

Specific comment 6: In discussion section, add concentration values for References 82, 83, and 72. If concentrations are not available, then low vs high concentrations can be mentioned. Simply stating associations is inadequate for readers.

Response 6: Thank you for your suggestions. We have added the specific concentrations of PFAS in Lum et al. and Fei et al. (page 11, lines 330 - 333). And highlighted that relatively low levels of PFAS might be negatively associated with endometriosis-related infertility in Wang et al. (page 11, lines 335 - 337).

Specific comment 7: Why was the justification used to stratify the data by only two specific age groups? What about other age group categories such as <25, 25-30, 30-35, 35-40, >40 ages for detailed analyses? If not, that could be added to the limitations.

Response 7: Thank you for your comments. The distribution of the prevalence of infertility by age groups of study participants is shown in Figure A. A total of 106 (13.45%) participants reported infertility. The infertility rates for the five age groups were: 8 (6.72%) for ages under 25, 9 (7.56%) for 25 to 30 years old, 14 (11.76%) for 30 to 35 years old, 18 (14.63%) for 35 to 40 years old, and 57 (18.51%) for age over 40. Since very few participants with infertility are younger than 35 years old, the statistical models might not be fitted. The confidence interval spans might be extremely wide. In addition, fertility rapidly declines after age 35[6], so setting the cutoff value for grouping standards in 35 years is reasonable. We have added this limitation to the discussion section (page 12, lines 370 - 372).

(Here is the Figure A)

Figure A. Cumulative numbers of five age groups, and the distribution of self-reported infertility by age group.

Reference:

  1. Li, Y.; Fletcher, T.; Mucs, D.; Scott, K.; Lindh, C. H.; Tallving, P.; Jakobsson, K., Half-lives of PFOS, PFHxS and PFOA after end of exposure to contaminated drinking water. Occupational and environmental medicine 2018, 75, (1), 46-51.
  2. Wang, W.; Zhou, W.; Wu, S.; Liang, F.; Li, Y.; Zhang, J.; Cui, L.; Feng, Y.; Wang, Y., Perfluoroalkyl substances exposure and risk of polycystic ovarian syndrome related infertility in Chinese women. Environmental pollution 2019, 247, 824-831.
  3. Kim, Y. R.; White, N.; Bräunig, J.; Vijayasarathy, S.; Mueller, J. F.; Knox, C. L.; Harden, F. A.; Pacella, R.; Toms, L.-M. L., Per-and poly-fluoroalkyl substances (PFASs) in follicular fluid from women experiencing infertility in Australia. Environmental Research 2020, 190, 109963.
  4. Luo, K.; Liu, X.; Zhou, W.; Nian, M.; Qiu, W.; Yang, Y.; Zhang, J., Preconception Exposure to Perfluoroalkyl and Polyfluoroalkyl Substances and Couple Fecundity: A Couple-Based Exploration. Environment International 2022, 107567.
  5. Wang, B.; Fu, J.; Gao, K.; Liu, Q.; Zhuang, L.; Zhang, G.; Long, M.; Na, J.; Ren, M.; Wang, A.; Liang, R.; Shen, G.; Li, Z.; Lu, Q., Early pregnancy loss: Do Per- and polyfluoroalkyl substances matter? Environment International 2021, 157, 106837.
  6. Harris, I. D.; Fronczak, C.; Roth, L.; Meacham, R. B., Fertility and the aging male. Reviews in urology 2011, 13, (4), e184.
